# Morphological and quantitative analysis of leukocytes in free-living Australian black flying foxes *(Pteropus alecto)*

Dale Hansen[1ᵒ]*, Brooklin E. Hunt[1ᵒ], Caylee A. Falvo[1ᵒ], Manuel Ruiz-Aravena[1], Maureen K. Kessler[2], Jane Hall[3,4], Paul Thompson[5], Karrie Rose[3], Devin N. Jones[1], Tamika J. Lunn[4], Adrienne S. Dale[6], Alison J. Peel[4], Raina K. Plowright[1], Bat One Health[¶]

**1** Department of Microbiology and Cell Biology, Montana State University, Bozeman, MT, United States of America, **2** Department of Ecology, Montana State University, Bozeman, MT, United States of America, **3** Australian Registry of Wildlife Health, Taronga Conservation Society Australia, Sydney, NSW, Australia, **4** Centre for Planetary Health and Food Security, Griffith University, Nathan, QLD, Australia, **5** Taronga Wildlife Hospital, Taronga Conservation Society Australia, Taronga Zoo, Sydney, NSW, Australia, **6** Department of Biological Sciences, Texas Tech University, Lubbock, TX, United States of America

ᵒ These authors contributed equally to this work.
¶ Membership of the BatOneHealth Consortium is provided in the Acknowledgments.
* dale.hansen@student.montana.edu

## Abstract

The black flying fox (*Pteropus alecto*) is a natural reservoir for Hendra virus, a paramyxovirus that causes fatal infections in humans and horses in Australia. Increased excretion of Hendra virus by flying foxes has been hypothesized to be associated with physiological or energetic stress in the reservoir hosts. The objective of this study was to explore the leukocyte profiles of wild-caught *P. alecto*, with a focus on describing the morphology of each cell type to facilitate identification for clinical purposes and future virus spillover research. To this end, we have created an atlas of images displaying the commonly observed morphological variations across each cell type. We provide quantitative and morphological information regarding the leukocyte profiles in bats captured at two roost sites located in Redcliffe and Toowoomba, Queensland, Australia, over the course of two years. We examined the morphology of leukocytes, platelets, and erythrocytes of *P. alecto* using cytochemical staining and characterization of blood films through light microscopy. Leukocyte profiles were broadly consistent with previous studies of *P. alecto* and other *Pteropus* species. A small proportion of individual samples presented evidence of hemoparasitic infection or leukocyte morphological traits that are relevant for future research on bat health, including unique large granular lymphocytes. Considering hematology is done by visual inspection of blood smears, examples of the varied cell morphologies are included as a visual guide. To the best of our knowledge, this study provides the first qualitative assessment of *P. alecto* leukocytes, as well as the first set of published hematology reference images for this species.

**Data Availability Statement:** All files are available from the Dryad database (DOI: 10.5061/dryad. gb5mkkws0).

**Funding:** RKP was funded by the DARPA PREEMPT program (https://www.preemptproject. org) Cooperative Agreement # D18AC00031, and the U.S. National Science Foundation (https://www. nsf.gov) (DEB-1716698). DH and BEH were funded by the Montana State University Undergraduate Scholars Program (https://www.montana.edu/usp/ ). BEH was also funded by the Montana State University McNairs Scholar Program (https://www. montana.edu/mcnair/) (Grant #P217A130148). RKP was also funded by the USDA National Institute of Food and Agriculture (https://nifa.usda. gov) (Hatch project 1015891). AJP was supported by an ARC DECRA fellowship (https://www.arc. gov.au/grants/discovery-program/discovery-early- career-researcher-award-decra) (DE190100710). The funders had no role in study design, data collection and analysis, decision to publish, or preparation of the manuscript.

**Competing interests:** The authors have declared that no competing interests exist.

## Introduction

Several bat-borne viruses are of human health concern when cross-species transmission events, or spillovers, occur. Presently, the majority of emerging human pathogens have an origin in animals, with many arising from wild mammalian species [1]. For a pathogen to spill over from its animal host into a human, multiple factors must align in space and time. This encompasses both ecological and epidemiological conditions, including changes in the health of the host that benefit pathogen shedding [2]. Thus, a reservoir host must shed the pathogen for it to potentially encounter a susceptible human to infect. As a result, spillover events are closely linked to the ability of a reservoir host's immune system to control infection, replication, and eventually pathogen shedding. In the case of bats, increased shedding of several viral species that infect them have been associated with periods of energetic or physiological stress [3–6]. The characterization of "health" is difficult in free-ranging wildlife, largely due to a lack of suitable metrics in wild animals with an unknown medical history. In this context, hematological analyses are frequently used as one of a suite of metrics to assess health, especially because of the role of white blood cells in the response of organisms to infections and other physiological stressors, which translates into changes in the numbers and proportions of different cell types [7]. If hematological changes have predictive power regarding infection outcomes, gaining a better understanding of bat hematology may help us to understand viral shedding, and therefore predict spillover events [8].

*Pteropus* spp. bats (colloquially known as flying foxes) are an important focus of bat-borne pathogen research [9]. Bat species within this genus are known or suspected reservoir hosts of several zoonoses, including Hendra virus, Nipah virus, Australian bat lyssavirus, and Menangle virus [6, 10–14]. In recent years in eastern Australia, flying foxes have increasingly occupied roosts within urban and peri-urban agricultural areas that offer predictable but low quality food resources [4]. These behavioral changes may facilitate contacts between bats and humans, and bats may also be experiencing poor health due to low food quality. As contact between flying fox populations, domestic animals, and humans increases, so does the risk of transmission of zoonotic infectious pathogens [3, 4, 6, 7]. One species of interest is *Pteropus alecto* (black flying fox) which play important ecological roles as pollinators and seed dispersers, and are endemic to Australia, Indonesia, and Papua New Guinea [15]. Despite the role of *P. alecto* as reservoirs of Hendra virus, relatively few studies have tried to characterize the physiological condition of clinically "healthy" individuals to serve as a baseline for health assessments. In the case of hematology, studies are generally conducted using automated cell counters. This approach, although highly efficient in terms of human hours and samples processed, can be impractical due to the logistics of using cytometers in remote field conditions, and the higher sample volume requirements for automated counting relative to slide preparation. Automated cell counters can also present errors associated with variability in cell types among species. In this context, manual assessment of cell morphology is required to establish a baseline of leukocyte characteristics. In this study, we aimed to address knowledge gaps regarding detailed morphological characterizations of the leukocytes of "healthy" black flying foxes. Here we provide an in-depth, microscopy-based characterization of *P. alecto* leukocytes, including morphological descriptions, reference photographs, and quantification of proportions of cell populations. The atlas of images provided in this study is aimed at facilitating the identification of *P. alecto* leukocytes in future studies and clinical settings.

## Materials and methods

### Animal ethics permits

This study was carried out in accordance with guidelines for animal care and handling under Griffith University Animal Ethics Committee (Approval ENV/10/16/AEC) and Montana State University IACUC Committee (#201750).

### Animal capture and sample collection

*P. alecto* were captured between June 2018 and July 2020 at two roost sites in Queensland, Australia: one in Redcliffe and one in Toowoomba (S1 Table). Redcliffe is a coastal suburb approximately 30 kilometers northeast of Brisbane. Toowoomba is a small inland city 700 m above sea level in the Great Dividing Range, approximately 120 kilometers west of Brisbane. These roost locations are spaced approximately 160 kilometers apart and are continuously occupied by *P. alecto* [16]. Bats were captured, examined, sampled, and released within their roost site.

Bats were captured pre-dawn in mist nets and anesthetized by a veterinarian or veterinarian-trained technician using 5% isoflurane in oxygen at 800 mL/min, followed by administration of 1.5% isoflurane in oxygen at the same rate once animals were fully anesthetized. Physical examination was conducted to identify age, sex, and body condition, as well as any macroscopic injuries or abnormalities. Age was estimated as juvenile (less than 1 year old), subadult (pre-reproductive, ~1–2 years old), and adult (greater than ~2 years old) based on morphometric measurements of forearm length, body mass, tooth appearance, and reproductive maturity [12, 17]. Distinction between adults and subadults was made based on penis and testes size and development in males, and pregnancy (via abdominal palpation) or evidence of past suckling (based on nipple protrusion and balding around nipples) in females.

All bats were marked by painting the claws on one hind limb with colored nail lacquer to identify recently captured bats and avoid resampling during the consecutive days of capture sessions. Bats captured from May 2019 onwards were also marked with a subcutaneous RFID Passive Integrated Transponder (PIT tag, or 'microchip'; ZD Tech Group, China) inserted between the scapulae. After sampling, each bat was monitored for recovery from anesthesia for at least 30 minutes; good grip ability and airway stability were confirmed before release.

Prior to PIT tag insertion, a maximum of 2.5 mL of blood was drawn from the cephalic or uropatagial vein, with samples not exceeding 0.6% of body mass. Blood smears were prepared on-site upon sample collection using blood drawn directly from the syringe, without the use of anticoagulant agents. Multiple smears were made for each bat when blood volume allowed. Smears were dried at ambient temperature and fixed in 100% methanol for three minutes. The samples were then stored at room temperature and out of UV light for up to two years until analysis at Montana State University (Import Permit No. 20200728-2504A).

### Hematological analysis

Blood smears were stained with the commercial Romanowsky stain variant DipQuik (Jorgensen Laboratories, Loveland, CO, 80538, USA). Shape, length, texture, and cell monolayer of each smear were examined for quality, and only medium and high-quality smears were analyzed. Smears with uneven distribution of leukocytes, damage or poor stain quality, and high numbers of reactive, unidentifiable leukocytes were not analyzed.

The morphological characteristics of leukocytes, erythrocytes, and thrombocytes in the monolayer were assessed using an AmScope E5 Biological Series microscope. Standard leukocyte differentials were performed manually by counting 100 different leukocytes in the monolayer of each smear. Up to three differential counts (up to 300 leukocytes total) were

performed on each smear to ensure at least 2/3 of the monolayer was examined. To avoid inconsistencies in identification of cells that could bias results, a single laboratorian conducted all smear examinations.

The diameters of 355 cells from a subset of 15 individuals sampled at both roost sites (n Redcliffe = 7; n Toowoomba = 8) were measured using an AmScope E5 series biological microscope and 5.1 MP USB3.0 Aptina Color CMOS Microscope Digital Camera. 134 erythrocytes, 80 neutrophils, 70 lymphocytes, 30 monocytes, and 28 eosinophils were measured, and cells which were extremely oblong, distorted, or abnormal in appearance were excluded. The camera software was calibrated at all magnification levels directly prior to measuring cells. The overall means, medians, and ranges for the diameter of each cell type were then calculated.

Reference images were taken at 600x or 1000x magnification on a Nikon Eclipse 80i microscope or an AmScope LED E5 Biological Series microscope equipped with a 5.1MP USB3.0 Aptina Color CMOS Microscope Digital Camera. Representative images of each cell type and repeatedly noted unique morphologies were taken from the highest quality smears to maximize image quality.

## Quantitative analysis

All data manipulation and visualizations were performed in R version 1.4.1106 using the packages tidyverse and ggplot2 [18–20]. Considering the low number of individual bats from which good quality smears were obtained, we opted to summarize the data providing interquartile ranges per sex and using only adult individuals (n = 134/134). Individual samples from both roosts were grouped to increase sample sizes. When differentials were performed on more than one smear from the same individual bat, the differentials were averaged so that each bat accounted for only one datapoint in the final dataset. Histograms were used to assess the normality of the data, and natural log-transformations in the form $\ln(x+1)$ were applied to the monocyte and eosinophil counts to facilitate visualizations by adjusting for skewness. Given the descriptive focus of this paper and the lack of absolute numbers of cells from which to draw robust statistical comparisons, no statistical analyses were performed.

## Results

The final dataset used for this study consisted of data collected from 134 adult *P. alecto* captured at Redcliffe and Toowoomba, Queensland, Australia at 16 different catching sessions between June 2018 and July 2020 (S1 Table). All bats used in our analysis appeared clinically healthy at the time of capture with no obvious indications of disease; however, minor injuries or abnormalities, including wing tears, abrasions, evidence of healed wounds, and mild dermatitis were observed on multiple individuals (n = 24; 13 females and 11 males). These individuals were excluded from calculations of the prevalence of morphological variations. A single individual was excluded from our analyses due to an exceptionally high neutrophil count. The anomaly was confirmed in two smears prepared from the individual and the reason for the elevated value remains unclear. An additional individual was also excluded from analysis due to pronounced emaciation.

### Morphological descriptions

**Neutrophils.** Neutrophils were the dominant leukocyte type, comprising over 50% of leukocytes in 92.72% of adult bats (n = 102/110) (Tables 1 and 2 and Fig 1). Neutrophils (Fig 2A–2H) were characterized by pale purple, pink, or grey cytoplasms and dark purple lobulated nuclei. There was high variability in the number of lobes observed, with numerous hyposegmented cells (less than three lobes) and hypersegmented cells (greater than four lobes)

**Table 1. Leukocyte profiles of *P. alecto* with no observed injuries.**

| Leukocyte Type | IQR | | Median | | Range | |
|---|---|---|---|---|---|---|
| | Female (n = 55) | Male (n = 55) | Female (n = 55) | Male (n = 55) | Female (n = 55) | Male (n = 55) |
| **Neutrophils** | 55.0–74.8 | 55.0–71.5 | 70.0 | 65.0 | 26.0–91.0 | 37.0–88.0 |
| **Lymphocytes** | 19.0–36.8 | 22.5–33.5 | 25.0 | 30.0 | 9.0–74.0 | 8.5–51.0 |
| **Eosinophils** | 0.0–3.5 | 0.3–6.0 | 1.0 | 2.0 | 0.0–28.0 | 0.0–26.0 |
| **Monocytes** | 2.0–6.0 | 2.0–5.3 | 4.0 | 4.0 | 0.0–10.0 | 0.0–14.0 |

Interquartile ranges, median values, and overall ranges for each observed leukocytic cell type per 100 cells counted for *Pteropus alecto* on which no injuries were observed.

observed (Fig 2G and 2H). Low numbers of activated neutrophils were observed throughout the sample set. Neutrophils were generally observed to be smaller in diameter than monocytes, but larger than lymphocytes (Table 3 and Fig 3). Neutrophils were primarily agranular with smooth cytoplasmic textural appearances, or hypogranular with slightly textured and darker staining cytoplasm. Granular neutrophils were observed in 3.63% of individuals (n = 4/110). In these cells, granules were large, round, purple to reddish pink in color, and dispersed throughout the cytoplasm at a low density (Fig 2C). Where present, granular neutrophils accounted for 1.4–4% of the total neutrophil population. Most granular neutrophils observed were banded or hyposegmented. Granular or agranular band neutrophils were observed in 34.54% of individuals (n = 38/110) (Fig 2A–2C).

## Lymphocytes

Lymphocytes were the dominant leukocyte type in 7.27% of individuals (n = 8/110) (Tables 1 and 2 and Fig 1). Lymphocytes displayed a high variability in size, shape, and color (Figs 2I–2P and 3 and Table 3). In general, lymphocytes were the smallest leukocyte type but varied in size, sometimes appearing similar in diameter to neutrophils (Table 3 and Fig 3). Most individuals had dimorphic or polymorphic lymphocyte populations. The cells were round to oval in shape, with a high nuclear to cytoplasmic ratio. Nucleus morphology was variable (round, oval, elliptical, or bean-shaped). A circular pale nucleolus was easily visible in a majority of larger lymphocytes. Binuclear lymphocytes were also observed in 2.72% of individuals (n = 3/110). These cells contained two clearly distinct nuclei of similar diameter (Fig 2N). A low number of reactive lymphocytes were regularly observed. Reactive lymphocytes were generally larger in size than non-reactive lymphocytes, showing distorted cytoplasmic and nuclear morphology with dark purple cytoplasmic edges (Fig 2M).

Unique large granular lymphocytes (LGLs) were observed and counted as part of the heterogenous lymphocyte population in 28.18% of individuals (n = 31/110). When observed, LGLs

**Table 2. Leukocyte profiles of *P. alecto* with observed mild/minor injuries.**

| Leukocyte Type | IQR | | Median | | Range | |
|---|---|---|---|---|---|---|
| | Female (n = 13) | Male (n = 11) | Female (n = 13) | Male (n = 11) | Female (n = 13) | Male (n = 11) |
| **Neutrophils** | 48.0–69.0 | 60.5–81.5 | 62.0 | 69.3 | 29.0–81.0 | 46.0–91.0 |
| **Lymphocytes** | 31.0–43.0 | 11.0–34.8 | 31.0 | 23.0 | 14.0–55.0 | 7.5–46 |
| **Eosinophils** | 0.0–2.0 | 0.0–1.0 | 1.0 | 0.3 | 0.0–12.5 | 0.0–5.5 |
| **Monocytes** | 3.0–7.0 | 2.8–7.0 | 5.0 | 5.0 | 0.5–9.0 | 0.0–9.0 |

Interquartile ranges, median values, and overall ranges for each observed leukocytic cell type per 100 cells counted for *Pteropus alecto* on which injuries were observed.

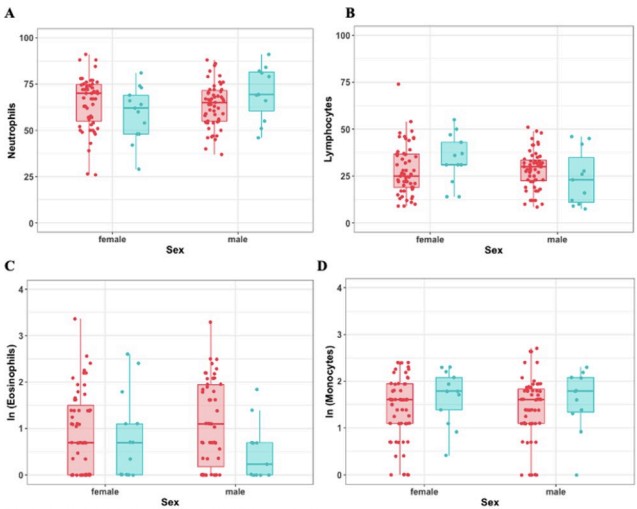

**Fig 1.** Box and whisker plots displaying the number of neutrophils (A), lymphocytes (B), eosinophils (C) and monocytes (D) observed per 100 cells counted in individuals on which injuries were noted (blue, n = 24) compared to individuals with no evident injuries (red, n = 110). Values of cell populations present in low abundances were log-transformed as ln(x+1) for visualization purposes.

comprised 3.8 to 13% of the general lymphocyte population. These lymphocytes were generally larger and lighter in color than the other lymphocytes typically observed and often had an oval

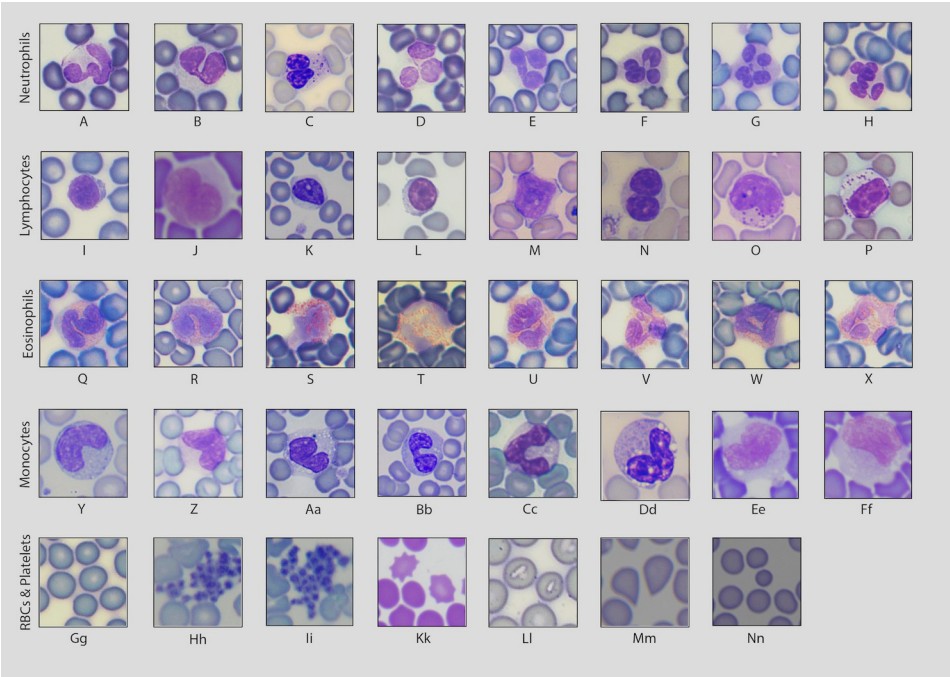

**Fig 2. Images taken at 600x or 1000x magnification depicting the variable morphologies of each leukocyte lineage as well as red blood cells and platelets.** (A, B) band neutrophils, (C) granular band neutrophil, (D-F) neutrophils, (G, H) hypersegmented neutrophils, (I-L) lymphocytes, (M) reactive lymphocyte, (N) binuclear lymphocyte, (O-P) large granular lymphocytes, (Q-R) band eosinophils, (S-X) eosinophils, (Y-Ff) mature monocytes, (Gg) erythrocytes, (Hh-Ii) platelet clumps, (Kk) echinocytes, (Ll) erythrocytes with low central pallor, (Mm) dacrocytes, (Nn) erythrocyte anisocytosis.

**Table 3. Cell diameters.**

| Cell Lineage | Mean (μm) | Median (μm) | SD (μm) | Range (μm) |
|---|---|---|---|---|
| Neutrophils (n = 80) | 12.15 | 12.23 | 1.04 | 11.00–13.06 |
| Lymphocytes (n = 70) | 9.44 | 9.53 | 1.36 | 8.38–10.17 |
| Eosinophils (n = 28) | 12.01 | 12.20 | 0.76 | 11.23–12.71 |
| Monocytes (n = 30) | 13.00 | 13.12 | 1.20 | 11.14–14.55 |
| Erythrocytes (n = 134) | 6.32 | 6.39 | 0.57 | 5.86–6.80 |

Mean values, median values, standard deviation, and overall ranges for the diameter of each observed cell type.

shape. The cytoplasm contained evenly distributed, reddish-pink, round-to-oval granules (Fig 2O and 2P). Granules were relatively large and visually striking but varied in size between individuals and were occasionally observed to have indistinct membranes.

## Eosinophils

Eosinophils were present in abundances that ranged up to 25% of the leukocyte population present in the samples (Tables 1 and 2 and Fig 1). Eosinophils (Fig 2Q–2X) presented similar diameter to neutrophils (Figs 2I-2P and 3 and Table 3) but smaller than monocytes (Figs 2Y–2Ff and 3 and Table 3). Eosinophil nuclei generally had three to four lobes, although some were hypersegmented (>4 lobes). Low numbers of band eosinophils were observed in 15.45% of individuals (n = 17/110) (Fig 2Q and 2R). The granules were relatively large and usually stained orange to pink, although a low number of granules were grey in color. Granules were often seen throughout only half to three-quarters of the cytoplasm, rather than equally distributed throughout the entirety of the cytoplasm (Fig 2S). Additionally, granules often stained lightly, making the differentiation of neutrophils and eosinophils challenging.

## Monocytes

Monocytes were observed to be highly variable in terms of overall cell size and color, both within and between individuals (Figs 2Y–2Ff and 3 and Table 3). These cells were generally observed to be larger than all other leukocytic cell types. Occasionally, monocytes were observed to be slightly smaller than neutrophils. Monocyte cytoplasm generally stained light to dark blue-grey with a classic "ground glass" textural appearance. Several monocytes were observed with round, dark purple cytoplasmic granules. The nuclei of mature monocytes were smooth and C-shaped, while immature monocytes (primarily promonocytes) showed an irregular and oval or round nucleus with a lumpy appearance. The chromatin patterns in most monocytes were often much smoother, finer, and lighter in color than that of neutrophils. However, chromatin patterns appeared to be affected by staining, as even mildly overstained smears showed monocytes with clumped chromatin patterns and slightly darkened, blue cytoplasmic borders. Low numbers of activated monocytes were observed throughout the sample set. Both vacuolated and non-vacuolated monocytes were observed, with vacuoles of highly variable sizes present in 58.18% of individuals (n = 64/110) (Fig 2Y–2Aa and 2Dd). 50.9% of individuals (n = 56/110) displayed both vacuolated and non-vacuolated monocytes.

## Basophils

Consistent with previous reports of healthy *P. alecto* presenting low numbers of basophils, these cells were not observed in the smears examined [21, 22].

## Erythrocytes

Erythrocytes stained with DipQuik generally displayed a normochromic coloration ranging from light to dark blue. Mild anisocytosis and polychromasia was commonly observed, and spherocytes, and cells with low central pallor were observed throughout the sample set (Fig 2Kk and 2Ll). Echinocytes (Fig 2Kk) were observed in 10% of individuals (n = 11/110) throughout our sample set. Dacrocytes (Fig 2Mm) were present in 48.1% individuals (n = 53/100). These cells were significantly smaller in diameter than all leukocytes observed (Table 3 and Fig 3).

## Platelets

Individual platelets were observed to be small and purple with a blurred, spiky plasma membrane; or as small, light grey vesicles containing small, round, dark purple granules. Platelets were observed either spread across smears or occasionally clinging to the sides of leukocytes. Platelet clumps were considered to be aggregations of multiple platelets and were observed in 59.9% of individuals (n = 56/110) (Fig 2Hh and 2Ii).

## Parasites

Potential hemoparasites were observed in the blood samples from one bat (Fig 4). Intracellular corpuscles consistent with the appearance of intraerythrocytic gametocytes (parasitic precursor cells) were observed in over 50 erythrocytes. These potential parasites varied in size, and a limited number of extracellular stages of the organism were also observed during the differential (Fig 4D). Generally, all stages of the organism showed distinguishable membranes containing mottled, granule-like components which often resembled beads on a string. Many of the potential intraerythrocytic parasites also showed one or two large inclusions (Fig 4C and 4D). An eosinophil count of 0 was observed on the differential cell count of this individual.

# Discussion

## Cell morphology

By conducting an extensive manual review of blood smears collected from 134 adult *P. alecto*, we were able to identify several trends within each observed leukocyte type. Although granular

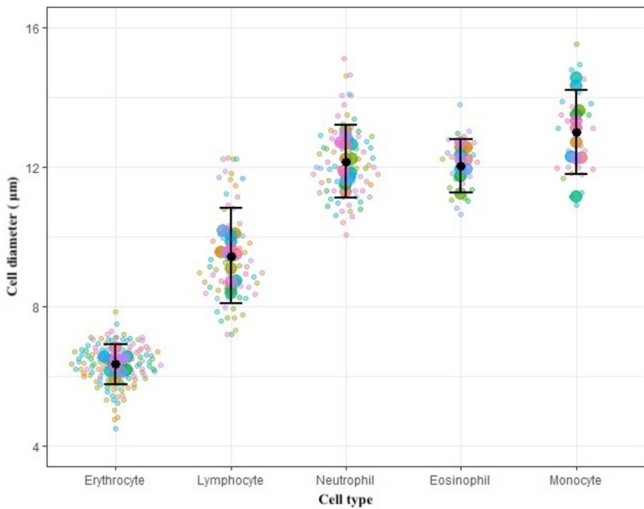

**Fig 3. Individual cell diameters observed in neutrophils (n = 80), lymphocytes (n = 70), eosinophils (n = 28), monocytes (n = 30), and erythrocytes (n = 134).** Each color represents an individual bat with larger dots depicting the mean for each cell type per bat. The error bars display the mean and 95% confidence intervals for each cell type.

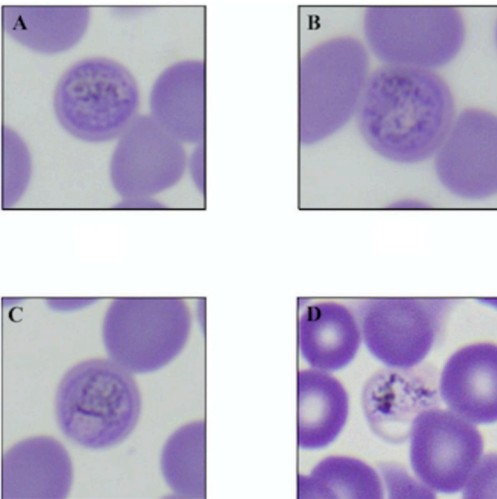

**Fig 4. Erythrocytes with signs of infection by possible *Hepatocystis* hemoparasites observed in a single individual captured at Redcliffe in December 2018.** Images taken at 1000x magnification.

neutrophils were observed in our samples, the granulation was likely not pathologically significant. Of the ten individuals in which this morphology was observed, two bats had evidence of lesions consistent with frostbite, and another had evidence of aural dermatitis. The other seven individuals had no noted abnormalities or injuries. Healthy neutrophils contain low numbers of primary granules, and although these granules typically do not stain vividly enough to be seen with light microscopy, the low numbers and even distribution of the granules observed in our samples is not suggestive of toxic granulation [23]. The variation in neutrophil lobulation observed in our sample set is consistent with the appearance of polymorphonuclear neutrophils (PMNs) in other bat species [24].

A low number of reactive lymphocytes were regularly observed, but this is likely not pathologically significant, as low numbers of reactive lymphocytes are regularly observed in blood smears from clinically healthy small mammals [23] (Fig 2M). Similarly, the monocytes observed in our samples were consistent with typical findings in other mammals. Monocytes were generally observed to be larger than all other leukocytic cell types and many contained vacuoles of varying sizes, which is consistent with observations made in other mammalian species [25]. Low numbers of activated monocytes were observed throughout the sample set, which is also typical for mammals [23].

Echinocytes (erythrocytes with crenated edges, Fig 2Kk) and dacrocytes (tear drop-shaped erythrocytes, Fig 2Mm) are two erythrocyte anomalies consistently observed in our samples which are most likely artifacts of sample collection or slide creation. Echinocyte formation in particular is frequently associated with prolonged smear storage time [26]. Given the low numbers of anomalous cells observed in individual smears and the possibility that the process of smear creation disrupted the morphology of the cells, these observations are likely not significant. The platelet clumps observed on our slides were also unlikely to be significant, as platelet clumps in blood smears are generally artifacts of venipuncture sample collection [27].

## Quantitative analysis

Neutrophils were observed to be the dominant cell type in the majority of samples, with lymphocytes rarely appearing as the dominant cell type on a given slide. Previous research has demonstrated that neutrophil dominance is a typical finding in many bat species,

including clinically normal *P. alecto* [21, 22, 28]. Our results are also consistent with findings of neutrophil dominance in other flying fox species, including the Rodriquez Island flying fox *(Pteropus rodricensis)*, Indian flying fox (*P. giganteus*), Christmas Island flying fox (*P. melanotus natalis)*, and the grey-headed flying fox (*P. poliocephalus*). However, the island flying fox *(P. hypomelanus)* and Malaysian flying fox *(P. vampyrus)* were observed to be lymphocyte-dominant [28–31].

No basophils were observed in our sample set, which is consistent with the low numbers of basophils typically reported in mammals, including other flying fox species [22, 24, 28, 30, 31]. The average numbers of eosinophils and monocytes observed on the smears in our study were also consistent with what has been reported previously [21, 22]. The low numbers of reactive neutrophils, monocytes, and lymphocytes observed across our sample set are consistent with findings in other mammalian species [23].

The large granular lymphocytes observed in 31 individuals (n = 31/110; none had observed injuries) were present across numerous samples from both sites. The distinctive morphological appearance we observed is consistent with descriptions of similar cells observed and described in multiple species of Neotropical bats [32]. We propose that these cells are natural killer (NK) cells or cytotoxic T cells, both of which have the appearance of normal lymphocytes but contain cytoplasmic granules [23]. However, additional cytochemical experiments are needed to confirm the identity of these cells. The apparent absence of reports of these cell types in other studies might result from the use of automated cell counts, which may not have differentiated these cells from other lymphocytes [21, 22]. By using a manual review of our smears, we were able to identify this consistently seen morphological anomaly as a noteworthy feature.

The intraerythrocytic hemoparasites observed in a single adult female from Redcliffe have a morphology that is consistent with *Hepatocystis* species. Although the appearance is generally consistent with reported *Hepatocystis* in other *Pteropus* species, molecular diagnostics would be needed to confirm taxonomic identity analysis that was not possible due to lack of appropriate samples [33]. This individual had an eosinophil count of zero; however, active endoparasitic infections are not always associated with eosinophilia at the time of infection [34]. Primary exposure to a parasitic organism generally results in delayed eosinophilia, which may manifest only after the parasitic organism dies, whereas subsequent exposures trigger intense, dramatic eosinophilia [23]. In our sample set, extremely elevated eosinophil counts (counts of 26% and 28% of observed cells, respectively), which may be evidence of recent parasitic pressure, were observed in one male and one female from Redcliffe captured in July and December 2019, respectively.

Although this study provides information about hematology in *P. alecto*, there are several limitations that should be considered. Due to the use of manual differential counts rather than flow cytometry or another automated counting technique, we are unable to report the total white blood cell concentration. While we were unable to state cell concentrations, we could report the proportions of leukocytic cells observed in our differential counts. The expense of automated methods and the need for fresh blood samples preclude the use of flow cytometry and other automated methods for population scale studies such as this. For population-level analyses, the relative proportions of leukocytic cells allows comparison of population hematological characteristics across large extents of space and time. In future studies, with larger sample sizes, such methods could be used to analyze temporal and more extensive spatial trends among reservoir host populations [35]. A previous study on this topic found seasonal differences in the neutrophil and lymphocyte counts which varied between males and females [21]. Finally, we focused on the hematology of adult members of this species, meaning that we were unable to report on the normal hematology of juvenile and sub-adult bats.

## Conclusion

We report the first qualitative assessment of the leukocytes, erythrocytes, and platelets of clinically healthy adult *P. alecto*. Our study also provides a set of values comparing the relative abundance of different leukocytes in adult *P. alecto*. Considering the time span of our sampling (2 years), our results are likely to capture the natural variability of cell populations in healthy, wild individuals. By examining our hematological data both qualitatively and quantitatively, we provide both an in-depth characterization of the normal hematological parameters for *P. alecto* and examine the variations in leukocyte ranges between males and females.

Due to their important ecological role as pollinators and seed dispersers, *Pteropodidae* bats are of significant interest to conservation efforts. Gaining a better understanding of the normal hematological profile of *P. alecto*, will facilitate efforts to monitor population health, which could contribute to the identification of populations under stressful conditions. Research of this nature will not only advance studies of *P. alecto* hematology and population health, but also inform studies of related species. A number of other *Pteropus* species are of immediate interest to the study of disease spillover, including *Pteropus medius*, *Pteropus lylei*, and *Pteropus vampyrus*, which have all been identified as Nipah virus reservoirs [11]. Reservoir health is intricately intertwined with the spillover of zoonotic disease. By gaining a better understanding of normal hematology in *Pteropus* bats, we can better monitor wildlife populations with the potential to shed zoonotic viruses.

## Supporting information

**S1 Table. Sampling effort across sites, sexes, and years.**
(DOCX)

## Acknowledgments

We acknowledge the Kabi Kabi and Yuggera Ugarapul people, who are the Traditional Custodians of the land upon which this work was conducted. The authors included in the Bat One Health consortium are as follows: Raina K. Plowright (Department of Microbiology and Cell Biology, Montana State University, Bozeman, MT, USA), Alison J. Peel (Centre for Planetary Health and Food Security, Griffith University, Brisbane, Queensland, Australia), Daniel Crowley (Department of Microbiology and Cell Biology, Montana State University, Bozeman, MT, USA), Wyatt Madden (Department of Microbiology and Cell Biology, Montana State University, Bozeman, MT, USA), Peter J. Hudson (Department of Biology, Center for Infectious Disease Dynamics, Pennsylvania State University, University Park, PA, USA), Peggy Eby (Centre for Planetary Health and Food Security, Griffith University, Brisbane, Queensland, Australia; School of Biological, Earth and Environmental Sciences, University of New South Wales, Sydney, New South Wales 2052, Australia), Andy Hoegh (Department of Mathematical Sciences, Montana State University, Bozeman, MT, USA), Hamish McCallum (Centre for Planetary Health and Food Security, Griffith University, Brisbane, Queensland, Australia; School of Environment and Science, Griffith University, Brisbane, Queensland, Australia), Liam P. McGuire (Department of Biological Sciences, Texas Tech University, Lubbock, TX, USA), Liam Chirio (Centre for Planetary Health and Food Security, Griffith University, Brisbane, Queensland, Australia), Mandy Allonby (Centre for Planetary Health and Food Security, Griffith University, Brisbane, Queensland, Australia), Rachael Smethurst (Centre for Planetary Health and Food Security, Griffith University, Brisbane, Queensland, Australia), Remy Brooks (Centre for Planetary Health and Food Security, Griffith University, Brisbane, Queensland, Australia), Kirk A. Silas (Wildlife Conservation Society, Health Program, Bronx, NY, USA),

Ticha Padgett-Stewart (Department of Microbiology and Immunology, Montana State University, Bozeman, MT, USA), Justine Scaccia, (Department of Microbiology and Immunology, Montana State University, Bozeman, MT, USA), and Denise Taimi Karkkainen (Environmental Futures Research Institute, Griffith University, Brisbane, Queensland, Australia; Biodiversity and Geosciences Program, Queensland Museum, South Brisbane, Queensland, Australia). The authors also acknowledge the support of Cara Parsons, Emma Glennon, Emily Stanford, Jessica Mitchell, Eloise Stephenson, Kerryn Parry-Jones, Anja Divljan, Cinthia Pietromonaco, and the Moreton Bay Regional Council and the Toowoomba Regional Council. KR acknowledges the support of The University of Sydney Institute for Infectious Diseases. Special thanks to Dr. Charlotte Hollinger, Dr. Dee McAloose, Dr. Siobhon Egan, Lauren Warner, Amelia Graves, and Lindsay Lee.

## Author Contributions

**Conceptualization:** Dale Hansen, Brooklin E. Hunt, Caylee A. Falvo, Maureen K. Kessler, Jane Hall, Paul Thompson, Karrie Rose, Raina K. Plowright.

**Formal analysis:** Dale Hansen, Caylee A. Falvo, Manuel Ruiz-Aravena.

**Funding acquisition:** Raina K. Plowright.

**Investigation:** Brooklin E. Hunt, Devin N. Jones, Tamika J. Lunn, Adrienne S. Dale, Alison J. Peel.

**Methodology:** Dale Hansen, Brooklin E. Hunt, Caylee A. Falvo, Manuel Ruiz-Aravena, Alison J. Peel.

**Resources:** Raina K. Plowright.

**Supervision:** Manuel Ruiz-Aravena, Raina K. Plowright.

**Visualization:** Dale Hansen, Caylee A. Falvo, Manuel Ruiz-Aravena.

**Writing – original draft:** Dale Hansen.

**Writing – review & editing:** Brooklin E. Hunt, Caylee A. Falvo, Manuel Ruiz-Aravena, Maureen K. Kessler, Jane Hall, Paul Thompson, Karrie Rose, Devin N. Jones, Tamika J. Lunn, Adrienne S. Dale, Alison J. Peel, Raina K. Plowright.

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
