## [Decision Letter · Decision Letter 0]

13 Dec 2021

PONE-D-21-35366Morphological and quantitative analysis of hemocytes in free-living Australian black flying foxes (Pteropus alecto)PLOS ONE

Dear Dr. Hansen,

Thank you for submitting your manuscript to PLOS ONE. After careful consideration, we feel that it has merit but does not fully meet PLOS ONE’s publication criteria as it currently stands. Therefore, we invite you to submit a revised version of the manuscript that addresses the points raised during the review process.

Both reviewers and myself agree that this is a well-written and helpful article for characterizing the hematological profiles of Australian flying foxes. Both reviewers provide a number of suggestions that will improve the manuscript. Most notably, reviewer 1 notes some discrepancies in how certain leukocytes are visualized/analyzed compared to other leukocytes as well as the need to explicitly analyze leukocyte data in relation to body condition and to identify the parasite gametocytes. Reviewer 2 also has also requested some additional analyses that would strengthen the paper. In relation to these comments, I had a few further comments for the authors: L80: Is there a reference the authors can cite for the ordinal body condition score? L124: Can the authors remind the reader what fraction of individual blood smears were from adults? L128: Note that the log transformation is not appropriate for proportion data, as proportion data are bound between 0-1 (see Warton & Hui 2011 Ecology for a full discussion). I suggest the authors consider logit transforming proportions, which also facilitates back-transforming the proportions to the true 0-1 boundary of the data. L159: Perhaps clarify here that “individuals” = “adult bats”. L160: Was a statistical test used (here and elsewhere) to compare leukocyte proportions between sexes and/or healthy/unhealthy bats? Or is this statement based only on data visualizations and raw summary statistics? I would suggest the authors consider the former as a more definitive way to make comparisons among groups. L275: Intraerythrocytic gametocytes of which parasites? It would help to identify the parasites to genus if possible, if not broader taxonomic level like order (e.g., haemosporidia).

We look forward to receiving your revised manuscript.

Kind regards,

Daniel Becker

Academic Editor

PLOS ONE

Journal Requirements:

We acknowledge the Kabi Kabi and Yuggera Ugarapul people, who are the Traditional Custodians of the land upon which this work was conducted. This study was funded by the DARPA PREEMPT program Cooperative Agreement # D18AC00031, and the U.S. National Science Foundation (DEB-1716698). DH and BH were funded by the Montana State University Undergraduate Scholars Program. BH was also funded by the Montana State University McNairs Scholar Program (Grant #P217A130148). RKP was also funded by the USDA National Institute of Food and Agriculture (Hatch project 1015891). AJP was supported by an ARC DECRA fellowship (DE190100710). KR acknowledges the support of The University of Sydney Institute for Infectious Diseases. Special thanks to Dr. Charlotte Hollinger, Dr. Dee McAloose, Dr. Siobhan Egan, Lauren Warner, Amelia Graves, and Lindsay Lee. 

RKP was funded by the DARPA PREEMPT program (https://www.preemptproject.org) Cooperative Agreement # D18AC00031, and the U.S. National Science Foundation (https://www.nsf.gov) (DEB-1716698). DH and BEH were funded by the Montana State University Undergraduate Scholars Program (https://www.montana.edu/usp/). BEH was also funded by the Montana State University McNairs Scholar Program (https://www.montana.edu/mcnair/) (Grant #P217A130148). RKP was also funded by the USDA National Institute of Food and Agriculture (https://nifa.usda.gov) (Hatch project 1015891). AJP was supported by an ARC DECRA fellowship (https://www.arc.gov.au/grants/discovery-program/discovery-early-career-researcher-award-decra) (DE190100710).The funders had no role in study design, data collection and analysis, decision to publish, or preparation of the manuscript.

5. One of the noted authors is a group or consortium Bat One Health. In addition to naming the author group, please list the individual authors and affiliations within this group in the acknowledgments section of your manuscript. Please also indicate clearly a lead author for this group along with a contact email address.

6. We note that Figures 2 and 3 in your submission contain copyrighted images. All PLOS content is published under the Creative Commons Attribution License (CC BY 4.0), which means that the manuscript, images, and Supporting Information files will be freely available online, and any third party is permitted to access, download, copy, distribute, and use these materials in any way, even commercially, with proper attribution. For more information, see our copyright guidelines: http://journals.plos.org/plosone/s/licenses-and-copyright.

a. You may seek permission from the original copyright holder of Figures 2 and 3 to publish the content specifically under the CC BY 4.0 license. 

Reviewers' comments:

Reviewer's Responses to Questions

**Comments to the Author**

1. Is the manuscript technically sound, and do the data support the conclusions?

Reviewer #1: Yes

Reviewer #2: Yes

2. Has the statistical analysis been performed appropriately and rigorously? 

Reviewer #1: Yes

Reviewer #2: Yes

3. Have the authors made all data underlying the findings in their manuscript fully available?

Reviewer #1: Yes

Reviewer #2: Yes

4. Is the manuscript presented in an intelligible fashion and written in standard English?

Reviewer #1: Yes

Reviewer #2: Yes

5. Review Comments to the Author

Reviewer #1: This paper reports the morphological and descriptive statistics of leukocyte parameters from 134 Bat flying foxes (Pteropus alecto) from Australia between 2018-2020. Pteropus species are known to be hosts of many zoonotic viruses of interest and therefore a descriptive study of this kind is incredibly important for understanding baseline health of these animals. Overall, this paper does a fantastic job of summarizing key statistics, including differences in animals with injuries and those deemed healthy, and provides a jumping off point for future studies to analyze blood smears. Overall, the paper is written quite well, and I have very few minor comments to improve the paper.

Reviewer #2: I was happy to read the manuscript entitled “Morphological and quantitative analysis of hemocytes in free-living Australian black flying foxes (Pteropus alecto)” written by Hansen and colleagues.

The authors analyzed the differential white blood cell counts (and red blood cells and platelets) of 134 Australian black flying foxes from blood smears via light microscopy and provide quantitative and morphological information on these – including an atlas of images. Although this method is time consuming, in my opinion has a tremendous value in comparison to more automated methods, especially for wildlife species. I very much like this study; despite the use of classic methods and a more descriptive study design, I personally think this is a nice manuscript, especially considering the recent interest of infectious biologists in bats, but also has value from a conservation physiology point of view.

Especially the atlas of images will be used not only by researchers but also by veterinarians, vet nurses – in a recent textbox about wild and exotic animal haematology, I could not find any photo or information on Chiroptera, despite being the second largest mammalian order. This manuscript definitively helps to fill this gap.

I have few comments, which hope can be easily addressed after a minor revision.

1. through the entire manuscript please replace hemocytes with white blood cells or leukocytes, as hemocytes refers to the cells of invertebrates

2. in the abstract it is written “…leukocyte morphological traits that are relevant for future research on bat health, especially in context of viral spillover and emergence”. I could not find these in the discussion. Would it be possible to rephrase this sentence, present and focus more the specific results?

3. introduction – “bat-borne pathogens”: are all these serious threat to human health, or some of them only? Again I think being more specific on viruses and/or intracellular pathogens, would help also to decrease the negative distinction of bats received lately due to their reservoir competence.

4. line 34 – “medical” history

5. lines 50 – 60: other drawbacks of automated methods are logistics (e.g. remote field conditions) and volume requirements (e.g. smaller sized species). These could be included in this section too.

6. The authors report and discuss the size of each cell type relative to others, which is indeed helpful when analyzing blood smears. However, I was wondering whether it would be possible to measure the size and thickness of each cell type and report these too?

7. figure 1 – please rearrange the order as differs between the figure and the order described in the legend

8. figure 1 and figure 2 – I would suggest to keep the logical order from the manuscript: neutrophils, lymphocytes, eosinophils and monocytes.

9. references - through the manuscript there were few sentences where I missed references supporting the statements. Could you please include these, as this would help the reader? See lines 320 (on neutrophil dominance in mammals); lines 356-364 (reference 19 is not the right one, please change it; LPS induced systemic inflammation causes neutrophilia in various bat species, including in Rousettus aegyptiacus; maybe some of these papers can be included here); lines 377-379 (indeed differential counts can allow comparison across larger scales. Reference missing – see Becker et al. 2019 Integr Comp Biol on common vampires).

10. I was surprised that the authors did not perform total white blood cell counts using standardized method with hemocytometer…It requires only small amount of blood and a field microscope. Why was this omitted? Although the differential counts alone have their values as discussed in this paper, still would be great if more studies would aim for total leukocyte counts too and not only via automated methods.

11. Besides the value of such study for disease research, I think also has importance for conservation (e.g. identifying anthropogenic stressors, species of conservation interest) – this aspect should be mentioned in the conclusion section.

6. PLOS authors have the option to publish the peer review history of their article (what does this mean?). If published, this will include your full peer review and any attached files.

Reviewer #1: No

Reviewer #2: No

---

## [Author Response · Author response to Decision Letter 0]

23 Feb 2022

Dear Dr. Becker,

RE: Revisions on manuscript PONE-D-21-35366 'Morphological and quantitative analysis of hemocytes in free-living Australian black flying foxes (Pteropus alecto)'

We appreciate the time and constructive critiques and comments from the reviewers and yourself. We have revised the manuscript accordingly.

Please, see below the specific responses to each of the comments. We present our responses, preceded by an “R”.

Editor comments:

L80: Is there a reference the authors can cite for the ordinal body condition score?

R: Per the recommendation of reviewer # 1, we have removed the body condition score statement from our methods. 

L124: Can the authors remind the reader what fraction of individual blood smears were from adults?

R: All the individual blood smears used in this study came from adults. We provide this information in lines 132-134 as follows: “Considering the low number of individual bats from which good quality smears were obtained, we opted to summarize the data providing inter-quartile ranges per sex and using only adult individuals (n=134/134).”.

L128: Note that the log transformation is not appropriate for proportion data, as proportion data are bound between 0-1 (see Warton & Hui 2011 Ecology for a full discussion). I suggest the authors consider logit transforming proportions, which also facilitates back-transforming the proportions to the true 0-1 boundary of the data.

R: We agree that the logit transformation suggested would have been appropriate to handle the data in a formal statistical analysis. In the specific case of the data presented, logit transformation would require extra manipulation of the data to account for zeros in the dataset, which were common for cell populations present in low abundances. We transformed the data only for visualization purposes and allow readers to back-transform the data if needed in a more intuitive and easier process than if it were done from a logit transformation. To make sure that values of zero in the transformed data presented do not confuse the readers, we have added the following phrasing in the figure legend for clarification in back-transformation: “Values of cell populations present in low abundances were log-transformed as ln(x+1) for visualization purposes”. 

L159: Perhaps clarify here that “individuals” = “adult bats”.

R: We have made this change. Please see lines 172-173 “Neutrophils were the dominant leukocyte type, comprising over 50% of leukocytes in 92.72% of adult bats (n=102/110).”

L160: Was a statistical test used (here and elsewhere) to compare leukocyte proportions between sexes and/or healthy/unhealthy bats? Or is this statement based only on data visualizations and raw summary statistics? I would suggest the authors consider the former as a more definitive way to make comparisons among groups.

R: Since the focus of our work was to present a reference of images and values for hematology of wild black flying foxes, we did not perform a formal statistical analysis, but rather focused on presenting the ranges of values. Though, we agree with the comment regarding formal comparisons. In this sense we consider that data on absolute numbers of cells which are not available for our dataset are needed to draw robust statistical comparisons. Acknowledging this caveat of our dataset, we adopted a conservative approach by presenting the dataset in a more descriptive way that could serve as baseline for future studies exploring the drivers of changes in leukocyte populations. 

L275: Intraerythrocytic gametocytes of which parasites? It would help to identify the parasites to genus if possible, if not broader taxonomic level like order (e.g., haemosporidia).

R: Unfortunately, no sample is available on which to run a PCR or similar test to confirm taxonomic identification of the potential hemoparasites. However, we have included a phrase referring to previous studies in which hemoparasites have been described. Please see lines 369-372 “Although the appearance is generally consistent with reported Hepatocystis in other Pteropus species, molecular diagnostic would be needed to confirm taxonomic identity [33]), analysis that was not possible due to lack of appropriate samples.” 

General editorial comments:

3. Thank you for stating the following in the Acknowledgments Section of your manuscript: We acknowledge the Kabi Kabi and Yuggera Ugarapul people, who are the Traditional Custodians of the land upon which this work was conducted. This study was funded by the DARPA PREEMPT program Cooperative Agreement # D18AC00031, and the U.S. National Science Foundation (DEB-1716698). DH and BH were funded by the Montana State University Undergraduate Scholars Program. BH was also funded by the Montana State University McNairs Scholar Program (Grant #P217A130148). RKP was also funded by the USDA National Institute of Food and Agriculture (Hatch project 1015891). AJP was supported by an ARC DECRA fellowship (DE190100710). KR acknowledges the support of The University of Sydney Institute for Infectious Diseases. Special thanks to Dr. Charlotte Hollinger, Dr. Dee McAloose, Dr. Siobhan Egan, Lauren Warner, Amelia Graves, and Lindsay Lee. 

We note that you have provided funding information that is not currently declared in your Funding Statement. However, funding information should not appear in the Acknowledgments section or other areas of your manuscript. We will only publish funding information present in the Funding Statement section of the online submission form. Please remove any funding-related text from the manuscript and let us know how you would like to update your Funding Statement. 

R: We have removed the funding-related text from the Acknowledgments. 

Currently, your Funding Statement reads as follows: RKP was funded by the DARPA PREEMPT program (https://www.preemptproject.org) Cooperative Agreement # D18AC00031, and the U.S. National Science Foundation (https://www.nsf.gov) (DEB-1716698). DH and BEH were funded by the Montana State University Undergraduate Scholars Program (https://www.montana.edu/usp/). BEH was also funded by the Montana State University McNairs Scholar Program (https://www.montana.edu/mcnair/) (Grant #P217A130148). RKP was also funded by the USDA National Institute of Food and Agriculture (https://nifa.usda.gov) (Hatch project 1015891). AJP was supported by an ARC DECRA fellowship (https://www.arc.gov.au/grants/discovery-program/discovery-early-career-researcher-award-decra) (DE190100710).The funders had no role in study design, data collection and analysis, decision to publish, or preparation of the manuscript.

R: Our amended funding statement reads as follows: “RKP was funded by the DARPA PREEMPT program (https://www.preemptproject.org) Cooperative Agreement # D18AC00031, and the U.S. National Science Foundation (https://www.nsf.gov) (DEB-1716698). DH and BEH were funded by the Montana State University Undergraduate Scholars Program (https://www.montana.edu/usp/). BEH was also funded by the Montana State University McNairs Scholar Program (https://www.montana.edu/mcnair/) (Grant #P217A130148). RKP was also funded by the USDA National Institute of Food and Agriculture (https://nifa.usda.gov) (Hatch project 1015891). AJP was supported by an ARC DECRA fellowship (https://www.arc.gov.au/grants/discovery-program/discovery-early-career-researcher-award-decra) (DE190100710). The funders had no role in study design, data collection and analysis, decision to publish, or preparation of the manuscript. The content of the information does not necessarily reflect the position or the policy of the U.S. government, and no official endorsement should be inferred.”

R: There are no changes we wish to make to the Data Availability statement. 

5. One of the noted authors is a group or consortium Bat One Health. In addition to naming the author group, please list the individual authors and affiliations within this group in the acknowledgments section of your manuscript. Please also indicate clearly a lead author for this group along with a contact email address.

R: The acknowledgments section of the manuscript has been updated to include the individual authors and affiliations within the BatOneHealth group. Please see lines 432-456 of the revised manuscript. The lead authors for the BatOneHealth group are Raina K. Plowright (raina.plowright@montana.edu) and Alison J. Peel (a.peel@griffith.edu.au). 

6. We note that Figures 2 and 3 in your submission contain copyrighted images. 

R: All images included in Figures 2 and 3 were taken by the authors for the purposes of this work. 

Reviewer 1:

Broad edits/comments:

It seems like you took the natural log of eosinophils and monocytes. Perhaps, I missed it but could you provide a short statement on why you did this for these two cell types and not neutrophils and lymphocytes.

R: We have added this statement. Please see lines 137-139 “Histograms were used to assess the normality of the data, and natural log-transformations in the form ln(x+1) were applied to the monocyte and eosinophil counts to facilitate visualizations by adjusting for skewness.”

You mention that you calculate body condition scores in your methods section (line 80). I suggest either include (1) an analysis that explicitly looks at the relationship between body condition and cell counts, (2) create a table that summarizes counts from bats that are in poor (i.e. below a threshold value) or good (i.e. above a threshold value), or (3) that you remove this statement from your methods.

R: We removed this statement from our methods. The body condition scores are based on a commonly used field assessment. We used the score to roughly filter individuals that may present clinical signs of unhealthy condition. We consider the scoring too coarse/inaccurate to include in a formal analysis and it may mislead conclusions. 

Minor edits:

Author list: “Bat One Health” is listed as an author with no affiliation details on the main manuscript document. Is this a mistake?

R: The full individual author and affiliation details can now be found in the acknowledgements section. Please see lines 432-456. 

Line 95: As they’re anesthetized it likely doesn’t matter as much, but could you state when the blood sample was taken? For example, prior to or after the PIT tag insertion.

R: The blood samples were taken prior to PIT tag insertion. We have added this information to the methods section. Please see lines 96-97, “Prior to PIT tag insertion, a maximum of 2.5 mL of blood was drawn from the cephalic or uropatagial vein, with samples not exceeding 0.6% of body mass.”

Line 122: Remove the extra space

R: We have made this change. Please see line 132. 

Line 275: Perhaps state what a gametocyte is? (e.g. precursor cells of a parasite)

R: We have defined gametocytes as “parasitic precursor cells” in lines 297-300. “Potential hemoparasites were observed in the blood samples from one bat (Fig. 3). Intracellular corpuscles consistent with the appearance of intraerythrocytic gametocytes (parasitic precursor cells) were observed in over 50 erythrocytes”

Figure 3 caption: Could you put in the caption which hemoparasite(s) you believe these to be?

R: We have updated the caption to include this information. Please see lines 309-310 “Fig 3: Erythrocytes with signs of infection by possible Hepatocystis hemoparasites observed in a single individual captured at Redcliffe in December 2018. Images taken at 1000x magnification.”

Reviewer 2:

1. through the entire manuscript please replace hemocytes with white blood cells or leukocytes, as hemocytes refers to the cells of invertebrates

R: We have made this change throughout the manuscript. 

2. in the abstract it is written “…leukocyte morphological traits that are relevant for future research on bat health, especially in context of viral spillover and emergence”. I could not find these in the discussion. Would it be possible to rephrase this sentence, present and focus more the specific results?

R: Considering this comment, we have rephrased the sentence. See lines 15-16. “...leukocyte morphological traits that are relevant for future research on bat health, including unique large granular lymphocytes.”

3. introduction – “bat-borne pathogens”: are all these serious threat to human health, or some of them only? Again I think being more specific on viruses and/or intracellular pathogens, would help also to decrease the negative distinction of bats received lately due to their reservoir competence.

R: We appreciate this comment and agree with the reviewer’s point. Accordingly, we have changed the language, see lines 22-23, “Several bat-borne viruses are of human health concern when cross-species transmission events, or spillovers, occur." In addition, we added information to the introduction regarding the ecological importance of the species in lines 48-53: “One species of interest is Pteropus Alecto (black flying fox) which plays important ecological roles as pollinators and seed dispersers, and are endemic to Australia, Indonesia, and Papua New Guinea [15]. Despite the role of P. alecto as reservoirs of Hendra virus, relatively few studies have tried to characterize the physiological condition of clinically “healthy” individuals to serve as baseline for health assessments.”

4. line 34 – “medical” history

R: Comment included in line 33: “…largely due to a lack of suitable metrics in wild animals whose medical history is unknown”.

5. lines 50 – 60: other drawbacks of automated methods are logistics (e.g. remote field conditions) and volume requirements (e.g. smaller sized species). These could be included in this section too.

R: We agree with this comment and have included it in the text. Please see lines 54-58. “This approach, although highly efficient in terms of human hours and samples processed, can be impractical due to the logistics of using cytometers in remote field conditions, and the higher sample volume requirements for automated counting relative to slide preparation. Importantly, automated cell counters can sometimes present errors associated with variability in cell types among species.” 

6. The authors report and discuss the size of each cell type relative to others, which is indeed helpful when analyzing blood smears. However, I was wondering whether it would be possible to measure the size and thickness of each cell type and report these too?

R: We have analyzed a subset of our samples to report the diameter of each cell type (see lines 117-123). We have added an additional table and figure to present intra and inter-individual variation of cell sizes. Please see Table 3: Cell Diameters, and figure 3.

7. figure 1 – please rearrange the order as differs between the figure and the order described in the legend

R: Figure 1 has been rearranged in the following order: neutrophils, lymphocytes, eosinophils, and monocytes. The legend has been updated accordingly. 

8. figure 1 and figure 2 – I would suggest to keep the logical order from the manuscript: neutrophils, lymphocytes, eosinophils and monocytes.

R: These changes have been made. Please see Figures 1 and 2. 

9. references - through the manuscript there were few sentences where I missed references supporting the statements. Could you please include these, as this would help the reader? See lines 320 (on neutrophil dominance in mammals); lines 356-364 (reference 19 is not the right one, please change it; LPS induced systemic inflammation causes neutrophilia in various bat species, including in Rousettus aegyptiacus; maybe some of these papers can be included here); lines 377-379 (indeed differential counts can allow comparison across larger scales. Reference missing – see Becker et al. 2019 Integr Comp Biol on common vampires).

R: The references in line 320 of the original manuscript were specific to bat species rather than mammals generally, so we have changed the language to reflect this. Please see lines 342-345: “Previous research has demonstrated that neutrophil dominance is a typical finding in many bat species, including clinically normal P. alecto (22,23,28).”. The change in references in lines 356-364 has been made, please see reference 21 in line 382-383 of the revised version. We have also included the recommended reference for lines401-405as reference 38. 

10. I was surprised that the authors did not perform total white blood cell counts using standardized method with hemocytometer…It requires only small amount of blood and a field microscope. Why was this omitted? Although the differential counts alone have their values as discussed in this paper, still would be great if more studies would aim for total leukocyte counts too and not only via automated methods.

R: We agree that this would have been a valuable analysis that would have contributed to a more quantitative approach (see our reply about statistical approach recommended by the editor), unfortunately this information was not collected due to time constraints for the field team to perform the analysis. To keep the fieldwork safe for the team, we tried to keep field days under 12 hours. We will consider field cell counts during future smaller studies.

11. Besides the value of such study for disease research, I think also has importance for conservation (e.g. identifying anthropogenic stressors, species of conservation interest) – this aspect should be mentioned in the conclusion section.

R: We appreciate the comment. We have updated both the introduction and the conclusion to include this information. Please see lines 46-53 “As contact between flying fox populations, domestic animals, and humans increases, so does the risk of transmission of zoonotic infectious pathogens (3,4,6,7). One species of interest is the black flying fox (Pteropus alecto). Bats of this species play an important ecological role as pollinators and seed dispersers, and are endemic to Australia, Indonesia, and Papua New Guinea (15). Despite the role of black flying foxes as reservoirs of Hendra virus, relatively few studies have tried to characterize the physiological condition of clinically “healthy” individuals to serve as baseline for health assessments.” 

And lines 417-420:

“Due to their important ecological role as pollinators and seed dispersers, Pteropus bats are of significant interest to conservation efforts. Gaining a better understanding of the normal hematological profile of P. alecto, will facilitate efforts to monitor population health, which could contribute to the identification of populations under stress.”

Thank you for your time and valuable comments.

Sincerely,

Dale Hansen, Brooklin E. Hunt, Caylee A. Falvo, Manuel Ruiz-Aravena, Maureen K. Kessler, Jane Hall, Paul Thompson, Karrie Rose, Devin N. Jones, Tamika J. Lunn, Adrienne S. Dale, Alison J. Peel, Raina K. Plowright

---

## [Decision Letter · Decision Letter 1]

14 Mar 2022

PONE-D-21-35366R1Morphological and quantitative analysis of leukocytes in free-living Australian black flying foxes (Pteropus alecto)PLOS ONE

Dear Dr. Hansen,

Thank you for submitting your manuscript to PLOS ONE. After careful consideration, we feel that it has merit but does not fully meet PLOS ONE’s publication criteria as it currently stands. Therefore, we invite you to submit a revised version of the manuscript that addresses the points raised during the review process.

The manuscript has now been seen by both original reviewers, who each agree that their prior concerns were addressed satisfactorily and that the manuscript will make a nice contribution to the literature on bat hematology and immunology more broadly. I agree with these assessments.

One of the conditions of publication in PLoS ONE is that conclusions are supported by data, and I would therefore ask a final set of edits by the authors. In the response letter, the authors note that the focus of the work was to present a reference of images and values for hematology of wild black flying foxes; because of this descriptive focus, the authors do not perform statistical analyses and instead present ranges (IQR) and medians of cell types. I don’t quite agree with the authors that you would need data on absolute numbers of cells to make such conclusions (you can analyze differences in the proportion of each leukocyte type), but descriptive analyses are perfectly fine for this journal. However, throughout the Results text, the authors do make inferential statements about how cell proportions differ between males and females and between apparently healthy and injured individuals (e.g., L174, “The interquartile ranges for neutrophils showed no major differences between males and females in apparently healthy individuals; however, in the group of bats with observed injuries, the interquartile ranges and medians were lower for females than males”). These statements are inferential, in that you are claiming a difference between the medians of two groups (this is the point of inferential statistics; e.g., a t-test or a non-parametric equivalent). I think these statements could mislead readers to infer comparisons that weren’t actually tested. Therefore, I would ask the authors to take one of two routes in this paper. On the one hand, you could clarify early in the Methods why you don’t perform formal statistical analyses, and then remove any of the inferential text from the Results. You could leave the tables and figures as they are, but not make any claims about median percent neutrophils, lymphocytes, eosinophils, etc being different between sexes or health status. On the other hand, you could retain these statements but support them with simple yet formal statistical tests (e.g., t-tests would be perfectly fine). Relevant parts of the Discussion (e.g., L380, 389) should then be modified or left as is, according to what the authors decide. Either option would fit well within PLoS ONE, but the authors should decide if they wish to infer sex- and health-related differences or leave the manuscript in a descriptive form.

Lastly, the authors must upload their raw data (i.e., individual-level data on leukocyte profiles) as a condition of acceptance.

We look forward to receiving your revised manuscript.

Kind regards,

Daniel Becker

Academic Editor

PLOS ONE

Journal Requirements:

Reviewers' comments:

Reviewer's Responses to Questions

**Comments to the Author**

1. If the authors have adequately addressed your comments raised in a previous round of review and you feel that this manuscript is now acceptable for publication, you may indicate that here to bypass the “Comments to the Author” section, enter your conflict of interest statement in the “Confidential to Editor” section, and submit your "Accept" recommendation.

Reviewer #1: All comments have been addressed

Reviewer #2: All comments have been addressed

2. Is the manuscript technically sound, and do the data support the conclusions?

Reviewer #1: Yes

Reviewer #2: Yes

3. Has the statistical analysis been performed appropriately and rigorously? 

Reviewer #1: Yes

Reviewer #2: Yes

4. Have the authors made all data underlying the findings in their manuscript fully available?

Reviewer #1: (No Response)

Reviewer #2: Yes

5. Is the manuscript presented in an intelligible fashion and written in standard English?

Reviewer #1: Yes

Reviewer #2: Yes

6. Review Comments to the Author

Reviewer #1: I believe that the authors did a good job at addressing all the reviewer's concerns and I believe it will make a great contribution to the field.

Reviewer #2: I would like to congratulate the authors for the revision they did on this manuscript. All my concerns and questions have been dealt with, I have no more suggestions.

7. PLOS authors have the option to publish the peer review history of their article (what does this mean?). If published, this will include your full peer review and any attached files.

Reviewer #1: No

Reviewer #2: No

---

## [Author Response · Author response to Decision Letter 1]

28 Apr 2022

Dear Dr. Becker,

RE: Revisions on manuscript PONE-D-21-35366 'Morphological and quantitative analysis of lymphocytes in free-living Australian black flying foxes (Pteropus alecto)'

We greatly appreciate the time you have taken to review this manuscript. We agree with the edits you have suggested, and after considering the two courses of revision you laid out in your review we have opted to keep the descriptive focus of this paper and remove any inferential language. This had led to changes in the results and discussion sections. Please see lines 176, 242, 272, 282, and 412 for the revisions made in these sections. We have additionally revised our methods section to clarify that we have not performed any formal statistical analyses. Please see lines 141-143. 

As a result of the changes made to the discussion section, we are no longer including the following three references:

35. Pacioni C, Robertson ID, Maxwell M, van Weenen J, Wayne AF. Hematologic Characteristics of the woylie (Bettongia penicillata ogilbyi). J Wildl Dis. 2013;49: 816–30.

36. Zuk M, McKean KA. Sex differences in parasite infections: Patterns and processes. Int J Parasitol. 1996;26: 1009–24. 

37. Christe P, Glaizot O, Evanno G, Bruyndonckx N, Devevey G, Yannic G, et al. Host sex and ectoparasites choice: preference for, and higher survival on female hosts. J Anim Ecol. 2007;76: 703–10.

To the best of our knowledge, the rest of the reference list is complete and correct, and we cite no retracted articles.

Thank you for your time and valuable comments.

Sincerely,

Dale Hansen, Brooklin E. Hunt, Caylee A. Falvo, Manuel Ruiz-Aravena, Maureen K. Kessler, Jane Hall, Paul Thompson, Karrie Rose, Devin N. Jones, Tamika J. Lunn, Adrienne S. Dale, Alison J. Peel, Raina K. Plowright

---

## [Editor Report · Decision Letter 2]

3 May 2022

Morphological and quantitative analysis of leukocytes in free-living Australian black flying foxes (Pteropus alecto)

PONE-D-21-35366R2

Dear Dr. Hansen,

We’re pleased to inform you that your manuscript has been judged scientifically suitable for publication and will be formally accepted for publication once it meets all outstanding technical requirements.

Kind regards,

Daniel Becker

Academic Editor

PLOS ONE
---

## [Editor Report · Acceptance letter]

10 May 2022

PONE-D-21-35366R2 

Morphological and quantitative analysis of leukocytes in free-living Australian black flying foxes *(Pteropus alecto)*

Dear Dr. Hansen:

I'm pleased to inform you that your manuscript has been deemed suitable for publication in PLOS ONE. Congratulations! Your manuscript is now with our production department. 

Kind regards, 

on behalf of

Dr. Daniel Becker 

Academic Editor

PLOS ONE